# Mental Health, Quality of Life, and Stigmatization in Danish Patients with Liver Disease

**DOI:** 10.3390/ijerph20085497

**Published:** 2023-04-13

**Authors:** Nadja Østberg, Birgitte Gade Jacobsen, Mette Munk Lauridsen, Lea Ladegaard Grønkjær

**Affiliations:** Department of Gastroenterology and Hepatology, University Hospital of Southern Denmark, 6700 Esbjerg, Denmark

**Keywords:** anxiety, depression, cirrhosis, hopelessness, liver disease, mental health, stigmatization, quality of life

## Abstract

The mental health of patients with liver diseases is often overlooked when assessing their overall health and planning care and treatment. The aim of this study was to assess anxiety, depression, hopelessness, quality of life, and the perception of stigmatization in a large cohort of patients with chronic liver disease of different aetiology and severity, as well as to identify predictors associated with mental health disorders. A total of 340 patients completed a survey assessing mental health using the Beck Anxiety Inventory, the Beck Hopelessness Scale, and the Major Depression Inventory. Quality of life was measured with the Chronic Liver Disease Questionnaire and the European Quality-of-Life visual analogue scale. To assess stigmatization, validated questions from the Danish Nationwide Survey of Patient Experiences were used. Predictors associated with anxiety, hopelessness, and depression were analysed using univariable and multivariable logistic regression analyses. Overall, 15% of the patients had moderate or severe anxiety, 3% had moderate or pronounced hopelessness, and 8% had moderate or severe depression. The prevalence of all three was highest in patients with cirrhosis and was associated with a low quality of life. More patients with cirrhosis had perceived stigmatization compared to patients with liver disease without cirrhosis, which affected their self-perception, and more than one-third of the patients refrained from telling others about their liver disease. The results emphasize the need for increased focus on mental health problems and awareness on preventing the discrimination of patients with liver disease.

## 1. Introduction

Liver disease is commonly caused by alcohol abuse, obesity (non-alcoholic steatohepatitis), viral hepatitis, and immune system abnormalities, and may progress to cirrhosis. Having a liver disease is often associated with serious health problems, hospitalization, and increased mortality, and the disease can have a major impact on the patient’s life due to complications and symptoms that may result in cognitive and physical impairment [1]. In addition, patients with liver diseases are at risk of being labelled regardless of the aetiology of the liver disease due to a lack of knowledge in the general population and misperceptions of liver diseases associated solely with alcohol and heavy drinking. Thus, patients with liver disease are vulnerable to mental health problems and reduced quality of life [2,3,4]. 

For decades, the mental morbidity of chronic diseases has been acknowledged. However, there has been little focus on mental health in patients with liver disease, which may have led to the underrecognition and undertreatment of mental health disorders [5]. Studies have shown that anxiety and depression account for the largest percentages of mental health disorders in patients with liver disease [6]. In addition, patients may experience vulnerability, for example, due to negative life events, problems such as financial and relationship issues, health concerns, or stigmatization, which may lead to feelings of hopelessness [7,8].

The emotional and psychological state of patients with liver disease may affect their self-management skills and result in reduced concordance with care and treatment. Accordingly, neglected mental health disorders are related to poor patient outcomes, including symptom progression, increased hospitalization burden, and mortality. Conversely, studies have found that patients with high levels of hope have a better prognosis and higher quality of life [7]. Therefore, the effective recognition and treatment of mental health disorders in patients with liver disease are important to plan liver disease care and treatment, reduce symptom burden and mortality, and secure optimal quality of life in patients with liver disease [3].

Different types of stigmatizations exist, such as public stigmatization, which involves negative or discriminatory attitudes toward others; self-stigmatization, which refers to the negative attitudes and shame that patients have about their own condition; and structural stigmatization, which involves policies of governments that intentionally or unintentionally limit opportunities for patient with liver disease [9]. The experience of stigmatization may impact patients negatively. Thus, studies have found an association between stigmatization and delayed disease recognition, timely help-seeking, and negative health outcomes, which may increase the overall burden of liver disease and cause mental health problems. Thus, increased knowledge are needed to lead action against the stigmatization of patients with liver disease [10,11].

The burden of mental health problems in Danish patients with liver disease is unknown. In addition, the patient’s experience of quality of life and perception of stigmatization have been scarcely studied [12]. Therefore, the aim of this study was to assess anxiety, depression, hopelessness, quality of life, and the perception of stigmatization in a cohort of Danish patients with liver disease of different aetiology and severity, as well as to identify predictors associated with mental health disorders.

## 2. Patients and Methods

A cross-sectional survey design to assess mental health, quality of life, and stigmatization in patients with liver disease, and the results are reported according to the Consensus-Based Checklist for Reporting of Survey Studies [13].

### 2.1. Patients

The participating patients comprised outpatients from the Department of Gastroenterology, University Hospital of Southern Denmark in Esbjerg with an established diagnosis of liver disease regardless of aetiology and severity based on the International Classification of Diseases (ICD-10). Exclusion criteria were an age below 18 years, lack of informed consent, and significant cognitive and/or linguistic barriers that hindered the completion of the questionnaire.

### 2.2. Development of the Survey

An online survey was developed by the authors to assess mental health, quality of life, and the perception of stigmatization. The survey consisted of individual, self-composed questions and validated questionnaires (see below). Prior to the data collection, two patients from the department’s patient panel, which is a panel of voluntary patients with the opportunity to influence and make suggestions for liver disease management and research, and three healthcare professionals with experience in developing questionnaires tested the survey to ensure that it was easily understandable and independent of reading capabilities. Based on the test, only minor linguistic changes were made. The survey was developed in Research Electronic Data Capture (REDCap), which is a secure, web-based software platform hosted by Open Patient Exploratory Network (OPEN), University of Southern Denmark [14]. Data on age, gender, diagnosis, and Model of End-Stage Liver Disease (MELD) score were manually collected from the medical chart of the patients and merged to the database. In the database, the patients were assigned a random ID number to secure anonymity.

### 2.3. Individual Survey Questions

The patients were asked to report information about their marital situation, social network, education, employment status, financial problems, smoking status, and loneliness. These questions were composed by the authors.

### 2.4. Alcohol Use

The Alcohol Use Disorders Inventory Test (AUDIT) was used to explore alcohol consumption, alcohol dependence, and alcohol-related problems. The instrument includes 10 items, and the total score ranges from 0 to 40. A higher score on the questionnaire indicates unhealthy alcohol use [15].

### 2.5. Assessment of Mental Health

The Beck Anxiety Inventory (BAI) was used to assess anxiety. The instrument consists of 21 items of descriptive statements on symptoms of anxiety with a 4-point Likert scale. The BAI score ranges from 0 to 63 and is classified as minimal anxiety (from 0 to 7), mild anxiety (from 8 to 15), moderate anxiety (from 16 to 25), or severe anxiety (from 30 to 63) [16].

The Beck Hopelessness Scale (BHS) was used to quantify hopelessness in the patients. It consists of 20 statements to measure positive or negative attitudes about the future, which are assigned a score of 0 or 1. The BHS score ranges from 0 to 20 and is classified as normal (from 0 to 3), mild hopelessness (from 4 to 8), moderate hopelessness (from 9 to 14), or pronounced hopelessness (over 14) [17].

The Major Depression Inventory (MDI) was used to assess depression. It consists of 12 items that cover the ICD-10 symptoms of depression. The MDI includes, in principle, 12 items, as item 8 and item 10 each have two sub-items, referred to as a and b, with a 6-point Likert scale. The total MDI score only covers 10 items as it is the highest score of either a or b in items 8 and 10 that are used. It ranges from 0 to 50 and is classified as no or doubtful depression (from 0 to 20), mild depression (from 21 to 25), moderate depression (from 26 to 30), or severe depression (from 31 to 50) [18].

### 2.6. Assessment of Quality of Life

The Chronic Liver Disease Questionnaire (CLDQ) and the European Quality-of-Life visual analogue scale (EQ VAS) from the European Quality-of-Life 5-Dimension 5-Level questionnaire (EQ-5D-5L) were used to assess patients’ quality of life. The CLDQ comprises 29 items encompassing six domains (abdominal symptoms, fatigue, systematic symptoms, activity, emotional function, and worry). Each item is rated on a 7-point Likert scale. A higher score on the questionnaire indicates minimum symptoms and a lower score indicates more pronounced symptoms. The CLDQ is specific regarding liver-related health issues [19]. The EQ VAS reports the overall current health, ranging from 0 (the worst health you can imagine) to 100 (the best health you can imagine) [20].

### 2.7. Assessment of Stigmatization

To assess stigmatization, eight validated questions from the Danish nationwide survey of patient perception were used [21]. The questions provided information on the patients’ perception of negative prejudices, other people perceiving the liver disease as self-inflicted due to alcohol, if the patients avoided telling others about the liver disease, and if the prejudices and negative attitudes affected the patients’ self-perceptions.

### 2.8. Data Collection Process

An invitation with a link to the electronic survey was distributed using patients’ digital and secure mailboxes, to which mail from Denmark’s entire healthcare is distributed. The first invitation was sent out in February 2021, and a reminder was sent out a month later. Two months later, the survey was sent out in paper format, including a return envelope for all the online non-responders. Patients who did not respond to this invitation were invited to the department to obtain assistance in completing the survey. The data collection ended in the spring of 2022.

### 2.9. Statistical Analyses

The data were analysed using Stata software version 17.0 (Stata Corp LP, College Station, TX, USA). Baseline demographic and disease characteristics and the scores from the questionnaires were summarised using descriptive statistics. Differences in the age, gender, liver disease diagnosis, and MELD score of responders and non-responders were explored. For subgroup analyses, the responding patients were divided into four groups (patients without cirrhosis, patients with cirrhosis, patients with non-alcoholic cirrhosis, and patients with alcohol-related cirrhosis) to analyse differences in mental health (assessed by the BAI, BHS, MDI), quality of life (assessed by the CLDQ and EQ VAS), and stigmatization. Due to non-normally distributed data, the Mann–Whitney test was performed for the analysis of variance, and the Chi-square test was used for comparisons of proportions.

Univariable and multivariable logistic regression analyses were used to identify predictors associated with mental health disorders (moderate or severe anxiety as opposed to no or mild anxiety, moderate or pronounced hopelessness as opposed to no or mild hopelessness, and moderate or severe depression as opposed to no or mild depression). The analyses were conducted separately for anxiety, hopelessness, and depression as outcome variables. The candidate predictor variables were age, male gender (yes/no), being in a relationship (yes/no), having a job (yes/no), having financial problems (yes/no), total AUDIT score, having cirrhosis (yes/no), MELD score, feeling lonely (yes/no), and CLDQ and EQ VAS scores. In addition to the liver disease measures, the predictor variables were selected from those reported to be associated with mental health disorders as described in previous studies [22]. Continuous variables were entered into the analyses in untransformed form.

## 3. Results

### 3.1. Patient Population

The survey was sent out to 581 patients and a total of 340 patients responded partially or fully, which gave a response rate of 59%. A flowchart of the data collection process is presented in Figure 1. Non-responders had a higher prevalence of alcoholic liver disease (10% versus 3%) and a higher MELD score (10 versus 9) (Table 1).

Baseline demographic and disease characteristics for the responders are presented in Table 2. The median age for all responders was 61 years, and 40% were women. Of the responders, 266 had liver disease without cirrhosis and 74 patients had cirrhosis. More patients with cirrhosis were male (64% versus 44%), fewer worked full time (16% versus 32%), more were smokers (35% versus 21%), and they had a higher AUDIT score (10, IQR 4–15 versus 5, IQR 2–7) compared to patient with liver disease without cirrhosis (Table 2). More patients with alcohol-related cirrhosis were single (36% versus 21%), were disability pensioners or chronically sick (34% versus 13% and 4% versus 0%), and were smokers (45% versus 18%) compared to patients with non-alcoholic cirrhosis.

### 3.2. Mental Health Problems

Anxiety: The median BAI score was 5.0 (IQR 2.0–11.0), and 15% of the patients reported either moderate or severe anxiety. Patients with cirrhosis had a higher median score compared to patients with liver disease without cirrhosis (6.0 versus 4.0). More patients with alcohol-related cirrhosis had severe anxiety compared to patients with non-alcoholic cirrhosis (19% versus 5%).

Hopelessness: A majority of the patients (92%) had a normal BHS score, and the median score was 0 with a range from 0 to 19. Patients with cirrhosis had a higher median score compared to patients without cirrhosis (1 versus 0). Patients with alcohol-related cirrhosis reported more mild hopelessness compared to patients with non-alcoholic cirrhosis (12% versus 3%).

Depression: The median MDI score was 7.0 (IQR 4.0–11.3), and 8% of the patients reported having moderate or severe depression. Patients with cirrhosis had a higher total MDI score (7.0, IQR 4.0–15.0) compared to patients with no cirrhosis (6.0, IQR 3.0–11.0). More patients with non-alcoholic cirrhosis had mild depression compared to patients with alcohol-related cirrhosis (11% versus 4%) (Table 3).

### 3.3. Quality of Life

The median CLDQ and EQ VAS scores for the patients were 5.3 (IQR 4.2–5.9) and 70 (IQR 50–81). Patients with cirrhosis had lower score in both the CLDQ (4.9, IQR 3.8–5.7 versus 5.4, IQR 4.3–5.9) and the EQ VAS (58, IQR 50–75 versus 73, IQR 50–83) compared to patients with liver disease without cirrhosis (Table 3).

### 3.4. Stigmatization

Overall, 5% of the patients had perceived negative prejudgements from healthcare professionals and 8% from relatives. In addition, 25% of patients with alcohol-related cirrhosis had perceived negative prejudgements from relatives, and the same fraction reported that prejudices and negative attitudes affected their self-perception. Patients with cirrhosis more often felt blamed by other people for having liver disease compared to patients with liver disease without cirrhosis (15% versus 46%), which affected their self-perception (23% versus 10%). Regardless of aetiology and severity, 32% of the patients avoided telling others about their liver disease. In addition, 44% of the patients with non-alcoholic cirrhosis perceived that other people had the misperception that their liver disease was caused by alcohol (Table 4).

### 3.5. Predictors of Mental Health Disorders

Having financial problems, the AUDIT score, the MELD score, feeling lonely, and quality of life were all associated with mental health disorders in the univariable analysis. However, only a low CLDQ score (low quality of life) was associated with moderate or severe anxiety (OR 0.13, 95% CI 0.05–0.30), moderate or pronounced hopelessness (OR 0.63, 95% CI 0.44–0.89), and moderate or severe depression (OR 0.18, 95% CI 0.09–0.37) in the multivariable analysis (Table 5).

## 4. Discussion

This study assessed mental health, quality of life, and the perception of stigmatization in a group of patients with liver disease to obtain a more in-depth understanding of how it is to live with liver disease.

The study found that 15% of the patients reported moderate or severe anxiety, 3% reported moderate or pronounced hopelessness, and 8% reported moderate or severe depression. The presence of anxiety, hopelessness, and depression were all associated with low quality of life. Only a small proportion of the patients had perceived prejudices from healthcare professionals. One-fourth of the patients and nearly half of the patients with cirrhosis had perceived other people blaming them for their liver disease, which affected patients’ self-perception and resulted in a large fraction of patients avoiding telling others about their liver disease. There were notable differences in the prevalence of mental health problems, low quality of life, and the perception of stigmatization, with higher proportions among patients with cirrhosis compared to liver disease patients without cirrhosis. These findings are in agreement with previous results, and a higher level of anxiety, hopelessness, and depression in patients with cirrhosis are likely given the risk of complications, hospitalizations, and the increased risk of morbidity and mortality [2].

Previous studies have recorded anxiety in 25–45% and depression in 29–72% of patients with liver disease, which is somewhat higher than in this study. Unfortunately, a direct comparison is difficult because of the use of different questionnaires and criteria for mental health problems. In addition, these studies have focused on the patient with advanced liver cirrhosis [6,23]. No study has been found in the literature that has examined the level of hopelessness in patients with liver disease.

In Denmark, the reported prevalence of anxiety and depression in the general population is 3% and 4%, respectively, which is the same as in other Nordic countries [19,23]. Although a comparison is impossible, the prevalence of mental health disorders in this liver disease population is higher. However, the reported prevalence of anxiety and depression in patients with chronic heart failure, chronic respiratory disease, and end-stage renal disease is between 10 and 52% and between 6 and 51%, respectively [24,25,26]. Thus, the results from this study indicate that patients with liver disease do not have a higher prevalence of anxiety and depression than other chronic patient groups. However, the results of the study highlight that healthcare professionals working with this patient group or patients with other chronic diseases should be alert to the prevalence of mental health disorders and focus on identifying patients in need of mental health counselling, for example, by systematically screening patients for mental health disorders in outpatient clinics. In addition, an integrated, multidisciplinary approach to accommodate patients with liver disease and co-occurring mental health disorders may be beneficial. Such an approach is supported by an abundance of research on the role of integrated care to improve concordance with care and treatment, quality of life, and reduce unhealthy lifestyles in patients with other chronic diseases [27,28].

There are several multifactorial factors to explain the association between liver disease and mental health disorders. Biologically, studies suggest that specific inflammatory and immune reactions in liver disease may mechanistically link liver disease and mental health disorders [29]. In addition, health behaviours such as alcohol abuse may lead to mental health problems and vice versa, as mental health problems may escalate alcohol use. Moreover, the severity of the liver disease has been associated with an increased risk of mental health disorders, as patients with cirrhosis may be cognitively and functionally impaired by the liver disease, resulting in fatigue, long-term stress, sleep disturbances, and social isolation. Stigmatization further increases the risk of mental health problems [30,31].

This study suggests that all patients, especially those with cirrhosis, would benefit from greater awareness of mental health problems. Ideally, such efforts should be evaluated in future intervention studies, which have not been conducted in patients with liver disease [5]. In other chronic patient groups, pharmacologic and psychosocial-based therapies have significantly improved quality of life, emotional and social functioning, and fatigue [32,33]. Moreover, psychosocial interventions have improved biological indicators of disease severity in diabetes and in ischemic heart disease [34,35].

Stigmatization can be associated with depression, lack of social support, and a decreased tendency to seek health care [1]. Patients with liver disease are often stigmatized, and this is particularly the case for alcohol-related liver diseases as the disease is not viewed as an illness but rather due to self-pity and weak will [36]. This patient group experiences more discrimination and severe stigmatization compared to other mental and medical disorders. In addition, population studies have shown that most responders blame those with alcohol-related liver disease for their disease in contrast to other mental disorders such as depression or schizophrenia. This may be due to a lack of knowledge on liver diseases and alcohol use disorders in the general population [37]. Stigmatization is not only a problem in alcoholic liver disease. A newer study found that perceived stigmatization is also common among patients with non-alcoholic fatty liver disease, is associated with impaired quality of life, and may be responsible for discrimination, shame, social isolation, and stereotypes, which may affect the human and social rights of affected patients [36]. Many of the patients choose not to tell others about their disease, likely because liver diseases, regardless of aetiology, are closely coupled to alcohol misuse in the view of the general population. In this study, this is illustrated by the fact that half of the patients who had a non-alcoholic liver disease had perceived other people believing the disease is due to alcohol. This may very well affect patients’ self-perception negatively. Healthcare professionals need to be aware of these perceptions and their influence on patients’ interactions and should reflect on stigmatization when counselling patients. One way of fighting stigmatization could be to encourage patients with a mental surplus to act as ambassadors for liver disease by making a point of telling people around them about their disease.

This study had several limitations. First, although the survey was sent online and physically, and patients were given the opportunity to obtain assistance to complete the survey to increase the response rate, the response rate was limited by nonresponse bias. However, our response rate was slightly higher than other online survey studies in this patient group [2,38]. Second, responders of the survey were most likely from the well-functioning half of the cirrhosis patients, and selection bias may have therefore influenced the results. However, the characteristics of respondents and non-respondents appeared similar overall. Third, variables such as anxiety, hopelessness, depression, quality of life, and the perception of stigmatization were estimated based on self-reporting questionnaires, which may result in a potential for over- or underreporting related symptoms. In addition, the questionnaire only assessed these variables at a single time point, so it cannot be determined if the mental health, quality of life, and stigmatization symptoms preceded the liver disease or would change over time. Moreover, other statistical analyses could have been performed to elucidate mental health problems in patients with liver disease. Finally, the results may not be generalizable to all patients with liver disease in Denmark. However, it is thought that these weaknesses are outweighed by several strengths, including a large sample size, in which it was possible to assess different levels of anxiety, depression, and hopelessness, and the use of validated questionnaires.

## 5. Conclusions

Mental health disorders were frequent in patients with liver disease and worst in patients with fully developed cirrhosis. In addition, patients perceived stigmatization regardless of aetiology and severity.

This study begins to address the gap of knowledge on mental health problems in patients with liver disease. Increased focus on mental health disorders should be considered to initiate an early integrated, multidisciplinary approach. In addition, future intervention studies with pharmacologic and psychosocial therapy must be performed in liver disease patients with mental health disorders to explore the effect on the quality of life and disease outcomes. For all liver disease patients, healthcare professionals should be supportive and should protect against stigmatization by encouraging initiatives such as increased information on liver diseases and inviting relatives to be part of the liver disease care and treatment plan. In addition, the findings call for increased awareness in the general population, patient associations, and policymakers in order to work toward preventing the discrimination and stigmatization of patients with liver disease.

## Figures and Tables

**Figure 1 ijerph-20-05497-f001:**
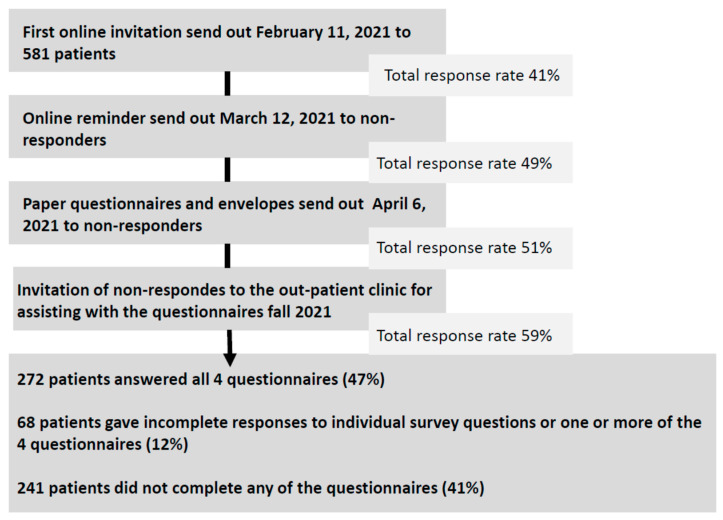
Survey invitation log and response rates.

**Table 1 ijerph-20-05497-t001:** Baseline characteristics of responders and non-responders.

	Responders (n = 340)	Non-Responders (n = 241)	*p*-Value
**Age—median (IQR)**	61 (52–69)	68 (59–76)	0.599
**Gender % female (Frequency)**	40% (136)	44% (106)	0.580
**Non-cirrhosis diagnosis (Frequency)**			
Non-alcoholic fatty liver disease	60% (159)	57% (78)	0.311
Autoimmune hepatitis	11% (29)	3% (4)	0.700
Primary Sclerosing cholangitis	20% (53)	26% (35)	0.877
Toxic liver disease	4% (11)	2% (3)	0.276
Alcoholic liver disease	3% (9)	10% (14)	**0.045**
Others	2% (5)	2% (3)	0.276
**Cirrhosis diagnosis (Frequency)**			
Alcohol-related cirrhosis	32% (24)	28% (29)	0.537
Non-alcohol cirrhosis	68% (50)	72% (75)	0.537
**MELD score—median (IQR)**	9 (7–12)	10 (8–16)	**0.016**

IQR, interquartile range. The Mann–Whitney and the Chi-square test was performed to explore the differences between responders and non-responders.

**Table 2 ijerph-20-05497-t002:** Baseline characteristics of all responders.

	All Responders	Liver Disease without Cirrhosis	Cirrhosis	*p*-Value	Non-Alcoholic Cirrhosis	Alcohol-Related Cirrhosis	*p*-Value
	(n = 340)	(n = 266)	(n = 74)		(n = 24)	(n = 50)	
**Age—median (IQR)**	61 (52–69)	61.5 (51–70)	64 (57–71)	0.423	66 (58–72)	64 (56–70)	0.481
**Gender—Female (Frequency)**	40% (136)	56% (149)	36% (27)	**0.004**	42% (10)	34% (17)	0.095
**Marital status (Frequency)**							
Single	31% (106)	31% (83)	31% (23)		21% (5)	36% (18)	**0.015**
In a relationship and living together	60% (205)	61% (163)	57% (42)	1	71% (17)	50% (25)	**0.002**
In a relationship not living together	3% (10)	3% (7)	4% (3)	0.565	4% (1)	4% (2)	1
Divorced	5% (18)	5% (12)	8% (6)	0.700	4 (1)	10% (5)	0.096
Other/unknown	0% (1)	0% (1)	0% (0)	0.389	0% (0)	0% (0)	
**Social network (Frequency)**							
0 persons in their network	2% (6)	2% (4)	3% (2)	0.651	4% (1)	1% (1)	0.174
1 person in their network	9% (28)	8% (19)	13% (9)	0.249	13% (3)	13% (6)	1
2 persons in their network	11% (34)	10% (24)	14% (10)	0.384	13% (3)	15% (7)	0.684
3 persons in their network	9% (27)	9% (22)	7% (5)	0.602	13% (3)	4% (2)	**0.022**
4 persons in their network	10% (30)	10% (24)	8% (6)	0.621	5% (0)	13% (6)	**0.048**
>4 persons in their network	58% (182)	60% (146)	51% (36)	0.200	58 (14)	47% (22)	0.119
Unknown/won’t answer	3% (8)	2% (5)	4% (3)	0.407	0% (0)	6% (3)	0.013
**Do you have children? (yes) (Frequency)**	82% (258)	82% (200)	82% (58)	0.911	88% (21)	79% (37)	0.268
**Years of formal education—median (IQR)**	10 (8–13)	10 (8–13)	10 (8–12)	0.513	11 (7–12)	10 (8–12)	0.598
**Employment status (Frequency)**							
Working full time	29% (98)	32% (86)	16% (12)	**0.008**	21% (5)	14% (7)	0.193
Reduced hours (<22–35 h)	9% (32)	11% (29)	4% (3)	0.066	13% (3)	0% (0)	**0.000**
Unemployed	2% (8)	2% (5)	3% (2)	0.651	4% (1)	2% (1)	0.407
Disability pensioner	14% (49)	11% (29)	27% (20)	**0.001**	13% (3)	34% (17)	**0.000**
Long-term sick leave	6% (19)	6% (17)	3% (2)	0.306	0% (0)	4% (2)	**0.043**
Retired	32% (109)	30% (81)	38% (28)	0.232	46% (11)	34% (17)	0.083
Student	2% (5)	2% (5)	1% (1)	0.561	4% (1)	0% (0)	**0.043**
Unknown	0% (0)	0% (0)	0% (0)		0% (0)	0% (0)	
Other	6% (20)	5% (1)	8% (6)	0.389	0% (0)	12% (6)	**0.000**
**Do you have financial problems?—yes (Frequency)**	15% (46)	12% (30)	23% (16)	0.080	8% (2)	30 (14)	0.513
**Do you feel lonely?—yes**	16%	13%	20%	0.100	22%	18%	0.604
**Do you smoke? (Frequency)**							
Yes	24% (66)	21% (44)	35% (22)	**0.027**	18% (4)	45% (18)	**0.000**
Never	39% (106)	43% (90)	26% (16)	**0.011**	41% (9)	18% (7)	**0.000**
Former smoking	37% (100)	36% (76)	39% (24)	0.661	42% (9)	37% (15)	0.469
**How often do you consume alcohol? (Frequency)**							
Never	30% (82)	25% (54)	45% (28)	**0.003**	43% (9)	46% (19)	0.669
Max once per month	23% (64)	25% (54)	16% (10)	0.115	24% (5)	12% (5)	**0.027**
2–4 times per month	25% (68)	29% (62)	10% (6)	**0.001**	14% (3)	7% (3)	0.106
2–3 times per month	13% (36)	14% (30)	10% (6)	0.384	5% (1)	12% (5)	0.076
4 times or more per month	9% (25)	6% (13)	19% (12)	**0.005**	14% (3)	23% (9)	0.101
**AUDIT score—median (IQR)**	7 (3–14)	5 (2–7)	10 (4–15)	**0.0125**	14 (7–18)	5 (3–15)	0.5416
**Non-cirrhosis diagnosis (Frequency)**							
Non-alcoholic fatty liver disease		60% (159)					
Autoimmune hepatitis		11% (29)					
Primary Sclerosing cholangitis		20% (53)					
Toxic liver disease		4% (11)					
Alcoholic liver disease		3% (9)					
Others		2% (5)					
**Cirrhosis diagnosis (Frequency)**							
Alcohol-related cirrhosis			32% (24)				
NASH-related cirrhosis			12% (9)				
Other and unspecified cirrhosis			18% (13)				
Cryptogenic			4% (3)				
Toxic cirrhosis			2% (1)				
Other			32% (24)				
**MELD—median (IQR)**	9 (7–12)		10 (8–15)		10 (8–13)	10 (8–15)	0.660

IQR, interquartile range. The Mann–Whitney and the Chi-square test was performed to explore the differences between patients with liver disease without cirrhosis versus patients with cirrhosis and patients with non-alcoholic cirrhosis versus patients with alcohol-related cirrhosis.

**Table 3 ijerph-20-05497-t003:** Assessment of mental health.

	All (n = 340)	Liver Disease without Cirrhosis (n = 266)	Cirrhosis (n = 74)	*p*-Value	Non-Alcoholic Cirrhosis (n = 24)	Alcohol-Related Cirrhosis (n = 50)	*p*-Value
**Beck Anxiety Inventory score -median (IQR)**	5.0 (2.0–11.0)	4.0 (1.0–9.0)	6.0 (2.5–17.0)	**0.006**	7.0 (2.0–15.0)	6.0 (3.0–21.5)	0.6816
Minimal anxiety (0–7)	64%	68%	54%	**0.029**	54%	60%	0.887
Mild anxiety (8–15)	21%	21%	21%	1	25%	13%	0.236
Moderate anxiety (16–25)	7%	7%	6%	0.602	9%	5%	0.082
Severe anxiety (26–63)	8%	4%	19%	**0.002**	12%	22%	**0.002**
**Beck Hopelessness Scale score—median (range)**	0 (0–19)	0 (0–19)	1 (0–19)	0.001	0 (0–19)	0 (0–12)	0.161
Normal (0–3)	92%	96%	86%	**0.037**	88%	84%	0.389
Mild hopelessness (4–8)	5%	2%	9%	**0.029**	3%	12%	**0.022**
Moderate hopelessness (9–14)	2%	1%	3%	0.329	3%	4%	0.836
Pronounced hopelessness (>14)	1%	1%	2%	0.561	6%	0%	**0.013**
**Major Depression Inventory score—median (IQR)**	7.0 (4.0–11.3)	6.0 (3.0–11.0)	7.0 (4.0–15.0)	0.0275	7.0 (4.0–19.8)	8.0 (4.0–14.5)	0.951
No or doubtful depression (<20)	88%	91%	81%	0.063	77%	84%	0.207
Mild depression (20–24)	4%	3%	7%	0.194	11%	4%	**0.024**
Moderate depression (25–29)	4%	3%	7%	0.700	4%	8%	0.471
Severe depression (>29)	4%	3%	5%	0.194	8%	4%	0.389
**Chronic Liver Disease Questionnaire—median (IQR)**	5.3 (4.2–5.9)	5.4 (4.3–5.9)	4.9 (3.8–5.7)	0.036	4.8 (3.9–5.8)	4.9 (3.8–5.7)	0.855
Abdominal symptoms	5.3 (4.0–6.0)	5.3 (4.0–6.0)	5.4 (4.3–6.3)	0.876	4.8 (3.2–6.3)	5.3 (4.3–6.3)	0.416
Fatigue	4.6 (3.0–5.6)	4.8 (3.0–5.8)	4.4 (3.0–5.4)	0.065	3.9 (2.8–5.4)	4.5 (3.2–5.2)	0.476
Systemic symptoms	5.4 (4.2–6.0)	5.4 (4.2–6.2)	5.2 (3.9–5.8)	**0.022**	4.6 (3.7–5.8)	5.4 (3.8–5.8)	0.395
Activity	5.6 (4.3–6.3)	5.6 (4.3–6.3)	5.3 (3.6–6.0)	**0.007**	5.7 (3.8–6.3)	5.3 (3.3–5.7)	0.292
Emotional symptoms	5.4 (4.3–6.1)	5.4 (4.3–6.1)	5.2 (3.9–6.0)	0.148	5.1 (3.9–6.2)	5.2 (3.8–6.0)	0.997
Worry	5.6 (4.4–6.4)	5.8 (4.4–6.6)	5.2 (4.0–6.3)	0.092	5.3 (3.7–6.4)	5.2 (4.4–6.3)	0.600
**European Quality-of-Life VAS score—median (IQR)**	70 (50–81)	73 (50–83)	58 (50–75)	**0.003**	62 (50–78)	56 (50–74)	0.401

IQR, interquartile range. The Mann–Whitney and the Chi-square test was performed to explore the differences between patients with liver disease without cirrhosis versus patients with cirrhosis and patients with non-alcoholic cirrhosis versus patients with alcohol-related cirrhosis.

**Table 4 ijerph-20-05497-t004:** Assessment of stigmatization.

	All (n = 340)	Liver Disease without Cirrhosis (n = 266)	Cirrhosis (n = 74)	*p*-Value	Non-Alcoholic Cirrhosis (n = 24)	Alcohol-Related Cirrhosis (n = 50)	*p*-Value
**Prejudices from healthcare professionals at outpatient clinic**	5%	3%	8%	0.083	0	11%	**0.000**
**Prejudices from healthcare professionals at in-patient care**	5%	4%	7%	0.243	3%	8%	0.335
**Prejudices from general practitioner**	5%	4%	4%	0.118	0	7%	**0.006**
**Prejudices from relatives**	8%	5%	17%	**0.000**	0	24%	**0.003**
**People blame you for your liver disease**	25%	15%	46%	**0.003**	43%	46%	0.377
**People think liver disease is due to alcohol**	33%	19%	63%	**0.000**	44%	NA	NA
**You avoid telling other people about your liver disease**	32%	30%	39%	0.095	34%	41%	0.533
**Prejudices and negative attitudes affect your self-perception**	14%	10%	23%	**0.013**	13%	26%	**0.023**

The Chi-square test was performed to explore the differences between patients with liver disease without cirrhosis versus patients with cirrhosis and patients with non-alcoholic cirrhosis versus patients with alcohol-related cirrhosis.

**Table 5 ijerph-20-05497-t005:** Univariable and multivariable logistic regression analyses of the association between the predictor variables and the outcome variables anxiety, hopelessness, and depression.

Variables	Univariable		Multivariable	
Moderate or severe anxiety				
	Odds ratio	95% CI	Odds ratio	95% CI
Age	0.99	0.97–1.01	0.73	0.42–1.26
Male (yes/no)	1.05	0.43–2.54	2.30	0.46–4.36
Relationship (yes/no)	1.29	0.75–2.22	1.64	0.02–3.78
Having a job (yes/no)	3.33	0.93–6.35	3.74	0.96–8.72
Having financial problems (yes/no)	**5.73**	**2.60–12.61**	4.25	0.98–9.77
AUDIT score	1.03	0.93–1.12	0.96	0.84–1.12
Having cirrhosis (yes/no)	1.54	0.86–2.75	2.40	0.97–5.57
MELD score	**1.10**	**1.01–1.23**	1.36	0.97–1.90
Feeling lonely (yes/no)	**3.68**	**1.75–7.76**	5.20	0.88–8.73
CLDQ score	**0.24**	**0.17–0.35**	**0.13**	**0.05–0.30**
EQ VAS score	**0.96**	**0.94–0.97**	1.01	0.93–1.09
Variables	Univariable		Multivariable	
Moderate or pronounced hopelessness				
	Odds ratio	95% CI	Odds ratio	95% CI
Age	1.03	0.97–1.09	1.18	0.81–1.73
Male (yes/no)	0.55	0.11–2.89	0.33	0.12–3.15
Relationship (yes/no)	1.89	0.53–6.67	1.16	0.46–6.97
Having a job (yes/no)	2.23	0.89–3.56	1.55	0.78–4.04
Having financial problems (yes/no)	2.76	0.68–11.09	2.50	0.54–9.64
AUDIT score	**1.28**	**1.14–1.42**	1.82	0.98–1.53
Having cirrhosis (yes/no)	3.19	0.84–12.17	1.85	0.44–7.71
MELD score	0.97	0.79–1.19	0.93	0.70–18.8
Feeling lonely (yes/no)	**3.82**	**1.04–14.04**	4.28	0.45–9.08
CLDQ score	**0.75**	**0.56–0.97**	**0.63**	**0.44–0.89**
EQ VAS score	0.97	0.95–1.00	1.00	0.94–1.06
Variables	Univariable		Multivariable	
Moderate or severe depression				
	Odds ratio	95% CI	Odds ratio	95% CI
Age	1.01	0.92–1.09	0.94	0.63–1.01
Male (yes/no)	0.31	0.08–1.12	0.17	0.02–9.78
Relationship (yes/no)	0.81	0.32–2.04	0.37	0.03–3.64
Having a job (yes/no)	5.09	0.16–22.31	2.21	0.16–2.84
Having financial problems (yes/no)	**7.62**	**3.07–18.89**	3.43	0.34–4.75
AUDIT score	1.05	0.87–1.18	1.18	0.92–1.50
Having cirrhosis (yes/no)	1.89	0.77–4.64	3.48	0.26–5.75
MELD score	1.12	0.98–1.27	1.19	0.80–1.78
Feeling lonely (yes/no)	**3.94**	**1.54–10.05**	3.22	0.64–9.14
CLDQ score	**0.33**	**0.23–0.49**	**0.18**	**0.09–0.37**
EQ VAS score	**0.96**	**0.93–0.97**	0.94	0.87–1.02

CI, confidence interval; MELD score, Model of end-stage liver disease; CLDQ, chronic liver disease questionnaire; EQ VAS score, European Quality-of-Life visual analogue scale. Bold fond *p* < 0.05 patients with moderate or severe anxiety vs. no or mild anxiety, patients with moderate or pronounced hopelessness vs. no or mild hopelessness, and patients with moderate or severe depression vs. no or mild depression.

## Data Availability

The dataset generated during and/or analysed during the current study is available from the corresponding author on reasonable request.

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
