# Peer review of "Mental Health, Quality of Life, and Stigmatization in Danish Patients with Liver Disease"

_ijerph, 2023, doi:10.3390/ijerph20085497_

Round 1

Reviewer 1 Report

This study was a cross-sectional survey included 340 patients with liver diseases and aimed to assess anxiety, depression, hopelessness, quality of life, and the perception of stigmatization. The results showed anxiety, depression, hopelessness was common in this population and all of them were sigficant predictor of quality of life. In addition, patients with cirrhosis were more likely to experience stigmatization compared to those without cirrhosis. Overall, this manuscript is well-written and has the potential to improve clinical practice if these mental problems could be addressed through appropriate treatments. The authors are to be congratulated on a very robust and interesting study.  

Author Response

Attached is response to Reviewer 1

Author Response

Attached is response to Reviewer 2

Reviewer 3 Report

The issue addressed is permanently topical.  The literature has presented the impact of hopelessness in different areas of life since 1968. Seligman, Abramson, Metalsky, are important representatives. 

The theoretical part (introduction) should be improved. In this form the theoretical foundation does not indicate clear scope and connections between concepts.

Aims need clarification.

Point 2.1. Patients - needs rigorous clarification on inclusion and exclusion criteria. If there were no other criteria the study is deeply flawed in terms of conceptualisation and construction of the experimental design.

Point 2.9. Statistical Analyses - clarifications on variables are needed. 

The processing of statistical data would have been much more thorough if parametric or nonparametric statistical tests had been used (depending on the data distribution). Frequency presentation is a low level of statistics. The data collected with the tools used (scales) would have allowed much more advanced processing and would (probably) have led to superior results.

The discussion section needs to be much improved.

It would be extremely useful to clearly present the novelty of the research as well as its practical (including clinical) usefulness, in other words, what is new.

Author Response

Attached is the response to Reviewer 3

Reviewer 4 Report

The manuscript “Mental Health: Quality of Life, and Stigmatization in Danish Patients with Liver Disease” is targeted at the relevant topic. Unfortunately, your research has to improve several aspects in order to be published. Your main problems are related to the fact that the structure of the introduction is clear. It is necessary to add more data and references. Moreover, in my humble opinion, it could be useful to describe in more detail the practical and theoretical implications of this research. It would be useful they contextualize better the contribution within the framework of the issue.

Author Response

Attached is the response to Reviewer 4

Reviewer 5 Report

The main objective of the article is "assess anxiety, depression, hopelessness, quality of life, and the perception of stigmatization in a large cohort of patients with chronic liver disease of different aetiologies and severity, as well as identify predictors associated with mental health disorders". This objective was beeen achieved. However it would be interesting to have performed a structural equation analysis with the significant predictor variables that appeared in the 3 models and the 3 criterion variables (depression, anxiety and hopelessness). It would be an interesting contribution, since the literature indicates that these mental disorders can be present together. I also suggest that the authors clarify the theoretical contributions of the study.

Author Response

Attached is response to Reviewer 5

Round 2

Reviewer 2 Report

Dear authors,

thank you for your understanding and acceptance of the proposal. I believe they have increased the quality of your valuable manuscript.

Best regards

Author Response

Attached is the reply for reviewer 2

Reviewer 3 Report

The adjustments are substantial. In this form the scientific level is much better.

Probably an in-depth analysis of the data analysis mechanisms (statistical methodology) would have led to a superior design.

I congratulate you for the effort to adjust the study.

Good luck in your scientific work.

Author Response

Attached is reply for reviewer 3
